# Non-Esterified Fatty Acid Induces ER Stress-Mediated Apoptosis via ROS/MAPK Signaling Pathway in Bovine Mammary Epithelial Cells

**DOI:** 10.3390/metabo12090803

**Published:** 2022-08-27

**Authors:** Yexiao Yan, Junpeng Huang, Changchao Huan, Lian Li, Chengmin Li

**Affiliations:** 1School of Biotechnology, Jiangsu University of Science and Technology, Zhenjiang 212100, China; 2Institute of Agricultural Science and Technology Development, College of Veterinary Medicine, Yangzhou University, Yangzhou 225009, China; 3College of Animal Science and Technology, Nanjing Agricultural University, Nanjing 210095, China

**Keywords:** non-esterified fatty acids, ER stress, apoptosis, bovine mammary epithelial cells

## Abstract

Elevated concentrations of non-esterified fatty acid (NEFA) induced by negative energy balance (NEB) during the transition period of dairy cows is known to be toxic for multiple bovine cell types. However, the effect of NEFA in bovine mammary epithelial cells (BMECs) remains unclear. The present study aimed to explore the role and molecular mechanism of NEFA in endoplasmic reticulum (ER) stress and the subsequent apoptosis in BMECs. The results showed that NEFA increased ER stress and activated the three unfolded protein response (UPR) signaling sub-pathways by upregulating the expression of GRP78, HSP70, XBP1, ATF6, phosphor-PERK, and phosphor-IRE1α. We also found that NEFA dose-dependently induced apoptosis in BMECs, as indicated by flow cytometry analysis and increased apoptotic gene expression. RNA-seq analysis revealed that NEFA induced apoptosis in BMECs, probably via the ATF4-CHOP axis. Mechanistically, our data showed that NEFA increased reactive oxygen species (ROS) levels, resulting in the activation of the MAPK signaling pathway. Moreover, quercetin, a well-known antioxidant, was found to alleviate ER stress-mediated apoptosis in NEFA-treated BMECs. Collectively, our results suggest that NEFA induces ER stress-mediated apoptosis, probably via the ROS/MAPK signaling pathway, as quercetin has been shown to alleviate ER stress-mediated apoptosis in NEFA-treated BMECs.

## 1. Introduction

The transition period from late pregnancy to early lactation is one of the most critical physiological stage in the productive life of high-yielding dairy cows, as nearly all the infectious and metabolic diseases occur during this period [1,2]. Decreased feed intake, postpartum milk production, and the requisite nutritional adaptations lead to a physiological state of negative energy balance (NEB) in transition cows. Severe NEB could cause subclinical ketosis and fatty liver disease, as well as affect the productive and reproductive performance throughout the lactation period [3,4]. Non-esterified fatty acid (NEFA) is the metabolite of fat mobilization initiated by NEB and therefore, it can be used as an indicator of NEB. Investigations have shown that an elevated concentration of NEFA is a crucial pathogenic factor which is probably related to the pathogenicity of some NEB-related metabolic disorders, such as ketosis or fatty liver, and high plasma levels of NEFA can lead to stress of the endoplasmic reticulum (ER) in the liver of periparturient dairy cows [5,6,7]. Moreover, NEFA treatment could activate ER stress and induce apoptotic damage in bovine granulosa cells and hepatocytes [8,9]. However, the ER response of bovine mammary epithelial cells (BMECs) to high concentrations of NEFA remains unclear.

The ER is the specific cellular site of appropriate protein biosynthesis, folding, assembly, modification, and trafficking. Various pathophysiological factors, such as nutrient deprivation, increase in general protein synthesis, altered redox balance, and calcium homeostatic changes, could cause the accumulation of unfolded/misfolded proteins in the ER, that is, ER stress [10,11]. During impaired homeostasis, an ER-specific adaptation program, known as the unfolded protein response (UPR), is activated to eliminate immature proteins, thus sustaining ER homeostasis. Three stress-sensor transmembrane proteins on the ER initiate the UPR signaling cascade: inositol requiring kinase 1 (IRE1/ERN1), PKR-like endoplasmic reticulum kinase (PERK, EIF2AK3), and activating transcription factor 6 (ATF6) [12,13]. In persistent or severe ER stress conditions, which are beyond the capabilities of the pro-survival ability of the UPR, cells undergo apoptosis, contributing to various pathological situations. Studies showed that ER stress-induced UPR was involved in pathophysiologic conditions mostly observed in the liver of transition cows, such as inflammation, the development of ketosis, or fatty liver [7,14]. Another work by Shinichi et al. revealed that UPR is associated with both differentiation and apoptosis of BMECs, thus affecting milk yield [15,16]. 

Transition dairy cows with NEB exhibit a high blood concentration of NEFA; ER stress and apoptosis in dairy cows may be associated with high levels of NEFA. Thus, we hypothesized that NEFA could induce mammary epithelial cell ER stress and apoptosis. The aim of this study was to explore the molecular mechanism of NEFA in inducing ER stress and apoptosis in BMECs.

## 2. Materials and Methods

### 2.1. Cell Culture and Treatments 

Bovine mammary epithelial cell lines (BMECs, MAC-T) were a gift from Dr. Youping Sun (Harvard University). Cells were grown to 80–90% confluence in Dulbecco’s Modified Eagle’s Medium (DMEM)/F12 medium (Gibco, Grand Island, NY, USA), supplemented with 10% fetal bovine serum (FBS; Zeta life, Menlo Park, CA, USA). Cells were cultured at 37 °C in an atmosphere of 90% humidity and 5% CO2. The medium was changed every 48 h. To determine the effect of NEFA on bovine mammary epithelial cells, cells were treated with 0, 0.3, 0.6, and 0.9 mM of NEFA for 3, 6, 9, and 12 h; 0 mM served as the control group [8,9]. Moreover, to investigate the effect of quercetin on NEFA-induced ER stress and apoptosis of BMECs in vitro, the cells were also divided into Con group (control, no treatment), Que group (quecertin: 10 μM, 24 h), NEFA group (NEFA: 0.9 mM, 6 h), and Que + NEFA group [17,18]. The concentration of stock NEFA solution was 50 mM, containing 21.75 mM oleic acid, 15.95 mM palmitic acid, 7.2 mM stearic acid, 2.65 mM palmitoleic acid, and 2.45 mM linoleic acid. 

### 2.2. Flow Cytometer Detection of Apoptosis

An Annexin V-FITC/PI Detection Kit (BD Biosciences, San Diego, CA, USA) was used for the determination of cell apoptosis. Following the manufacturer’s instructions, the cells were treated with 0, 0.3, 0.6, and 0.9 mM of NEFA for 6 h, and then were stained with Annexin V-FITC and PI at room temperature for 25 min. Apoptosis was measured with a FACS Calibur (BD Biosciences, Bedford, MA, USA) flow cytometer (FCM). The data analysis was performed using Flowjo software version 10.0.7 (Becton, Dickinson and Company, Franklin Lakes, JD, USA).

### 2.3. Transmission Electron Microscopy (TEM)

At room temperature, BMECs were treated with 0.9 mM NEFA for 6 h and then harvested and fixed in 2.5% glutaraldehyde. The fixed cells were postfixed in 1% osmium tetroxide, dehydrated using a graduated ethanol series (30, 50, 70, 80, 90, and 100%) for 10 min each, embedded in Epon (Energy Beam Sciences, Agawam, MA, USA), sliced into ultrathin sections (50–60 nm) using a Leica EM UC6 ultramicrotome (Leica Microsystems, Wetzlar, Germany), and then stained with 3% uranyl acetate and lead citrate. The ultrathin sections were observed under an H7500 transmission electron microscope (Hitachi, Tokyo, Japan).

### 2.4. Intracellular Reactive Oxygen Species Measurement

The BMECs were seeded into 6-well plates at 1 × 10^6^ cells and then treated with NEFA for 6 h. A ROS-sensitive probe DCFH-DA (2,7-Dichlorodihydrofluorescein Diacetate) is used for the measurement of intracellular ROS level (Nanjing Jiancheng, China). The method was based on the DCFH-DA, which can enter cells freely through living cell membranes, and hydrolyzed into DCFH by intracellular esterases, and DCFH is easily oxidized to DCF (dichlorofluorescein), which is a strong green fluorescent substance that can be measured at excitation and emission wavelengths of 500 and 525 nm. BMECs, both control and NEFA-treated, are incubated with 10 µM of DCFH-DA in the dark for 30 min at 37 °C, then resuspended in PBS and immediately analyzed by a fluorescence microscope (Zeiss LSM 700 META (Olympus, Tokyo, Japan)).

### 2.5. Real-Time Quantitative PCR Analysis

Cells were treated with various concentrations (0, 0.3, 0.6, and 0.9 mM) of NEFA for 3, 6, 9, and 12 h for RT-qPCR. Total RNA was extracted using TRIzol reagent (Invitrogen, Carlsbad, CA, USA). The purity and quantity of total RNA were detected using a NanoDrop 1000 spectrophotometer (Thermo Scientific, Wilmington, DE, USA). DNase I and the Prime Script RT Master Mix kit (TaKaRa, Otsu, Japan) were used for DNA removal and cDNA synthesis, respectively. Real-time quantitative PCR was performed using standard protocols on an Applied Biosystem 7500 HT Sequence detection system using SYBR ^®^Premix Ex Taq™ (TaKaRa, Otsu, Japan). The PCR system consisted of 10 μL SYBR Premix Ex Taq, 2 μL cDNA, 0.4 μL ROX Reference Dye II, 0.4 μL of each primer (10 µM), and 6.8 μL double distilled water in a total volume of 20 μL. The PCR program consisted of one cycle at 95 °C for 30 s, 40 cycles at 95 °C for 5 s, and 60 °C for 34 s, with fluorescence signal collection at 60 °C. The primer sequences were shown in Table 1. Gene expression data were normalized to that of glyceraldehyde-3-phosphate dehydrogenase (GAPDH) by employing an optimized comparative Ct (2^−ΔΔCt^) value method. 

### 2.6. RNA-Seq and Transcriptome Analysis

Three samples were selected from the control and 0.9 mM NEFA treatment group for acquiring transcriptome sequences. Total RNA of NEFA-treated and untreated cells were extracted by TRIzol reagent (Invitrogen, Carlsbad, CA, USA); and RNA quality was assessed on an Agilent 2100 Bioanalyzer (Agilent Technologies, Palo Alto, CA, USA) and checked using RNase free agarose gel electrophoresis. The cDNA fragments were purified with a QiaQuick PCR extraction kit (Qiagen, Venlo, The Netherlands), end repaired, poly(A) supplemented, and ligated to Illumina sequencing adapters. The ligation products were size-selected by agarose gel electrophoresis, PCR amplified, and sequenced using Illumina HiSeq2500 by Gene Denovo Biotechnology Co., Ltd. (Guangzhou, China). Reads were filtered by fastp (v 0.18.0) and mapped to the reference genome (Bos Taurus, assembly ARS-UCD1.2 (GCA_002263795.2), http://asia.ensembl.org/Bos_taurus/Info/Index, accessed on 5 May 2021) using HISAT2. 2.4. The mapped reads of each sample were assembled by using StringTie (v1.3.1) in a reference-based approach. RNAs differential expression analysis was performed by DESeq2 R package. The genes with the parameter of false discovery rate (FDR) below 0.05 and absolute fold change ≥2 were considered differentially expressed genes (DEGS). Gene ontology (GO) and KEGG functional enrichment analyses of DEGs were performed by hypergeometric distribution tests. A *p*-value ≤ 0.05 indicated significantly enriched GO and KEGG terms. GO and KEGG pathway enrichment analysis of differentially expressed genes was transformed into a Bubble plot and a heatmap. The raw data were submitted to the National Center for Biotechnology Information (NCBI) under BioProject accession number PRJNA869860.

### 2.7. Western Blot Analysis

The BMECs were grown in a 6-well plate and treated with various NEFA concentrations (0, 0.3, 0.6, and 0.9 mM) for 6 h. Whole BMECs protein lysates were centrifuged at 15,000 rpm for 15 min at 4 °C, and supernatants were quantified for total protein using a BCA protein assay kit (Beyotime, Beijing, China), according to the manufacturer’s instructions. Proteins were separated by SDS-PAGE and transferred to a PVDF membrane. The membranes were probed with the following primary antibodies: rabbit anti-GRP78, Bcl2, Caspase3, CHOP, IRE1 (1:1000, Proteintech, Chicago, IL, USA), rabbit anti-HSP70, PERK, phosphor IRE1, XBP1, ATF6, p38 MAPK, phosphor-p38 MAPK, ERK, phosphor-ERK, JNK and phosphor-JNK (1:1000, ABclonal, Boston, MA, USA), rabbit anti-ATF4, phosphor PERK (1:500, Bioss, Beijing, China), rabbit anti-Bax, and Tubulin (1:1000, Bioworld, Louis Park, MN, USA). The blots were incubated with HRP-conjugated secondary antibodies, and the signals were detected by enhanced chemiluminescence (ECL) Western blot detection reagents (Pierce, Rockford, IL, USA). Immunoblots were scanned and densitometry was performed using ImageJ software (v 1.48, National Institutes of Health, Bethesda, MD, USA).

### 2.8. Statistical Analysis

Statistical analyses were performed using GraphPad Prism 6.01 software (GraphPad Software Inc., San Diego, CA, USA). All values are shown as the mean ± standard error of the mean (SEM) from three repeats of each experiment, run in triplicate. One-way analysis of variance (ANOVA), followed by Tukey’s test, was used for more than two groups, and a Student’s t test was used to compare the two sets of data. *p* < 0.05 was considered a significant difference.

## 3. Results

### 3.1. NEFA Induced ER Stress in BMECs

The results showed that the expression of GRP78 was significantly increased after treatment with 0.3 mM NEFA for 6 and 12 h, and 0.9 mM NEFA for 3 h, and this increase was more pronounced after stimulation with various NEFA concentrations (0.6 and 0.9 mM) for 6, 9, and 12 h (Figure 1A). Moreover, consistent with the expression of GRP78 mRNA, the expression of HSP70, an ER-associated chaperone protein, was also upregulated after treatment with 0.9 mM NEFA for 3 h, and this increase was more pronounced after stimulation with various concentrations (0.6 and 0.9 mM) of NEFA for 6 and 9 h; no significant differences were found after cells were stimulated with NEFA for 12 h (Figure 1B). Western blot analysis revealed that the NEFA (treatment 6 h) upregulated the GRP78 and HSP70 protein expression in a dose-dependent fashion (Figure 1C). Moreoever, ER stress was also discovered and confirmed under conventional TEM, as indicated by dilated endoplasmic reticulum (Figure 1D).

### 3.2. NEFA Activated the Three UPR Signaling Pathways in BMECs

We also analyzed the effects of NEFA on the UPR signaling pathway triggered by ER stress. As shown in Figure 2, the PERK, IRE1, and ATF6 mRNA expression were significantly increased after treatment with various concentrations (0.3, 0.6, and 0.9 mM) of NEFA for 6 h (Figure 2A,C,D); and the mRNA expression of PERK, eIF2α, IRE1α, XBP1, and ATF6 were significantly upregulated after stimulation with 0.9 mM NEFA for 9 h (Figure 2A–E). Western blot analysis revealed that the NEFA (treatment 6 h) significantly increased the P-PERK, P-IRE1α, XBP1, and ATF6 protein expression (Figure 2F). Taken together, these results suggest that NEFA induces ER stress and activates the three UPR signaling pathways in BMECs.

### 3.3. NEFA Induced Apoptosis in BMECs

The apoptotic level of BMECs was detected to analyze the cytotoxic effect of NEFA. Quantitative real-time PCR analysis showed that the mRNA expression of Caspase 3 and Bax were increased after treatment with 0.3 mM NEFA for 12 h, and 0.9 mM NEFA for 3 h, and this increase was more pronounced after stimulation with various concentrations (0.6 and 0.9 mM) of NEFA for 6, 9, and 12 h (Figure 3A,B). Bcl2, which is also related to apoptosis signaling, was significantly decreased after stimulation with 0.9 mM of NEFA for 6, 9, and 12 h (Figure 3C). Western blot results showed that NEFA (treatment 6 h) significantly increased the cleaved-caspase3 and Bax/Bcl2 protein levels in the BMECs (Figure 3D,E). At the same time, flow cytometry analysis using annexin V FITC/PI double staining showed that NEFA led to cell apoptosis in a dose-dependent manner (Figure 3F,G).

### 3.4. NEFA Induced Apoptosis Probably via ATF4-CHOP Axis

To further explore the mechanism of NEFA-induced apoptosis in BMECs, the NEFA treatment-activated genes, with a focus on ER stress sensors and the subsequent apoptotic responses, were analyzed by RNA sequencing. As shown in Figure 4A, genes with NEFA treatment were significantly enriched in the biological processes related to apoptosis. Furthermore, regarding the molecular function (MF) of GO enrichment analysis, we observed that misfolded and unfolded protein binding related to ER stress were significantly enriched, and the significant cellular component included the CHOP-ATF4, CHOP-ATF3 complex. Of note, the enrichment results of commonly regulated genes included the ER-stress sensors ATF4, CHOP(DDIT3), GRP78(HSPA5), and TRIB3; the expression modes of these genes in each sequencing sample are shown in Figure 4B. qPCR verification is shown in Figure 4C; the qPCR results were consistent with the RNA-seq results. Western blot results showed that NEFA (treatment 6 h) significantly increased the ATF4 and CHOP(DDIT3) protein levels in the BMECs (Figure 4D). Protein–protein interaction analysis determined possible interactions between CHOP(DDIT3) and ATF4 (Figure 4E). Collectively, these data suggested that NEFA induced apoptosis in BMECs under ER stress, probably via the ATF4-CHOP axis.

### 3.5. NEFA Caused Accumulation of ROS in BMECs

NEFA can serve as energy molecules that are oxidized in the mitochondria, accompanied by a significant increase in ROS production [8,9]. Therefore, the intracellular ROS levels in NEFA-treated BMECs were determined by DCFH-DA staining. Results showed that ROS signals were upregulated by NEFA treatment in a concentration-dependent manner (Figure 5).

### 3.6. MAPK Signaling Pathway Was Involved in NEFA-Induced Apoptosis

Based on the KEGG analysis enrichment results, the MAPK signaling was the most significantly enriched pathway upon NEFA treatment (Figure 6A). We also quantified MAPK pathway-related protein levels by Western blotting after NEFA stimulation in BMECs. As shown in Figure 6B, compared to the control, the phosphorylated p38 MAPK, c-Jun N-terminal kinase (JNK) and extracellular signal-regulated kinase (ERK) expression in the NEFA treatment groups were increased to some extent, indicating that NEFA treatment markedly activated the MAPK signaling pathway.

### 3.7. Quercetin Alleviated ER Stress-Mediated Apoptosis in NEFA Treated BMECs

To ascertain the protective role of quercetin in ER stress-mediated apoptosis in NEFA treated BMECs, cells were pre-treated with quercetin (10 μM) for 24 h, then treated with 0.9 mM NEFA for 6 h. The results showed that the protein levels of GRP78, HSP70, ATF4, and CHOP were significantly decreased in the quercetin-treated group as compared with the NEFA group (Figure 7A). Furthermore, quercetin also significantly suppressed NEFA-induced apoptosis, as indicated by flow cytometry analysis and the reduced Bax/Bcl2 protein levels in the BMECs (Figure 7B,C).

## 4. Discussion

Elevated plasma concentration of NEFA induced by NEB is a crucial pathogenic factor in perinatal cows. The close association between NEFA and hepatic metabolic and molecular adaptations has been widely studied, whereas little is known about the molecular response of mammary cells to high concentrations of NEFA [19]. Here, we illustrated the effects and common molecular events of NEFA in BMECs. We reported that NEFA treatment (1) induced ER stress and activated the three UPR branches, (2) induced the subsequent apoptosis in BMECs, probably via ATF4-CHOP axis, (3) caused accumulation of ROS in a concentration-dependent manner, and (4) activated the MAPK signaling pathway. The addition of the antioxidant quercetin could alleviate ER stress-mediated apoptosis. These findings are relevant to the molecular mechanisms of ER stress-mediated apoptosis under NEFA stimulation in BMECs.

Emerging evidence has indicated that ER stress occurs in the transition period of high-yielding dairy cows who face the challenge of NEB, and that it is closely connected with the development of infectious and metabolic diseases [7,20,21,22]. Moreover, various studies have reported that high levels of NEFA can be a very crucial stimulus of ER stress in perinatal cows [5]. Our study found that NEFA treatment led to a significant increase in the expression of ER stress markers in BMECs (Figure 1), which was consistent with the results obtained in bovine hepatocytes and granulosa cells [8,9,23]. Thus, we confirmed that NEFA stimulation could trigger ER stress in BMECs. UPR, caused by ER stress, can function as an adaptive cellular response to alleviate ER stress and regain ER proteostasis [24]. In the present study, NEFA treatment activated the three branches of UPR (Figure 2), indicating that UPR was effectively triggered in response to the excessive values of NEFA. 

However, if the adaptive UPR cannot restore ER homeostasis, persistent ER stress culminates in apoptotic cell death [25,26,27]. Here, we found that a significant increase in cell apoptosis was induced by NEFA in BMECs, as indicated by flow cytometry analysis and an increase in apoptotic gene expression (Figure 3); the results are consistent with previous research on the cytotoxic role of NEFA in ER stress-induced apoptosis of bovine granulosa cells and hepatocytes [8,9]. To gain insight into the mechanisms of ER stress-mediated apoptosis, we performed RNA-seq to define the transcriptional states in BMECs treated with NEFA; the sequencing and verification results showed that the expression of ER stress markers ATF4 and CHOP was significantly upregulated (Figure 4). Numerous studies have shown that the ATF4-CHOP pathway is the vital component involved in ER stress-induced apoptosis [28,29,30]. ATF4, a transcription factor, induces the transcription of genes required to relieve ER stress. Furthermore, ATF4 also induces its downstream target, CHOP(DDIT3), with noted roles in the induction of apoptosis during ER stress [31,32]. As a proapoptotic transcription factor, CHOP contributes to apoptosis by inducing the pro-apoptotic factors, such as Bim13, and inhibiting the expression of anti-apoptotic Bcl2. Therefore, we speculated that NEFA induced ER stress-mediated apoptosis, probably via the ATF4-CHOP axis.

ROS, functioning as signaling molecules, play a critical role in the regulation of cell death in response to different stimuli [33,34,35]. A series of studies have shown that ROS accumulation could initiate apoptotic pathways, thereby resulting in the death of bovine mammary epithelial cells [36,37,38]. Our data also provide evidence that the trigger of apoptosis was induced by NEFA via a mechanism that involves excessive ROS production (Figure 5). Interestingly, there is strong evidence that ROS-mediated apoptosis is dependent on the ATF4-CHOP axis [39,40]. Furthermore, ROS are also well known to exert their effect on activating the mitogen-activated protein kinase (MAPK) cascade pathways, therefore acting upstream of MAPKs. The classical MAPK pathway family consists of the JNK, p38 MAPK, and ERK, which mediate many cellular responses, including apoptosis, proliferation, survival, and differentiation under different stresses [41]. Consistent with previous research showing that high levels of NEFAs were able to induce apoptosis by activating the MAPK signaling pathway [8], our results also showed that NEFA could stimulate the MAPK pathway, as indicated by an increased phosphorylation of p38/JNK/ERK (Figure 6). Taken together, our data provide evidence that NEFA activate the ROS/MAPK signaling pathway to induce ER stress-mediated apoptosis in BMECs.

Quercetin, a member of the flavonoid family, is a plant-derived metabolite that has been reported to present a wide range of biological functions, including anti-oxidant, anti-inflammatory, and anti-apoptotic activities [42,43]. Furthermore, quercetin has been reported to modulate ER stress and was found to modulate cell proliferation and apoptosis in various types of cells [17,18,43,44,45]. Consistent with those reported in the articles, our results also showed that quercetin could alleviate ER stress-mediated apoptosis in NEFA treated BMECs, as evidenced by the downregulated expression of ER stress and apoptosis markers (Figure 7). Further in vivo study using a clinically relevant NEB model and clinical studies are required to assess the potential applications of quercetin as a novel therapeutic approach for NEB in transition cows.

## 5. Conclusions

Collectively, our study demonstrates that high levels of NEFAs induce ER stress and promote apoptosis in BMECs; the intracellular ROS and MAPK signaling pathway are probably involved in ER stress-mediated apoptosis triggered by NEFA. Moreover, we found that treatment with quercetin in vitro could attenuate ER stress-mediated apoptosis in NEFA-treated BMECs (Figure 8). The findings from the present study will promote new exploration on the transition cow’s adaptive physiologic mechanisms to prevent or alleviate negative energy balance.

## Figures and Tables

**Figure 1 metabolites-12-00803-f001:**
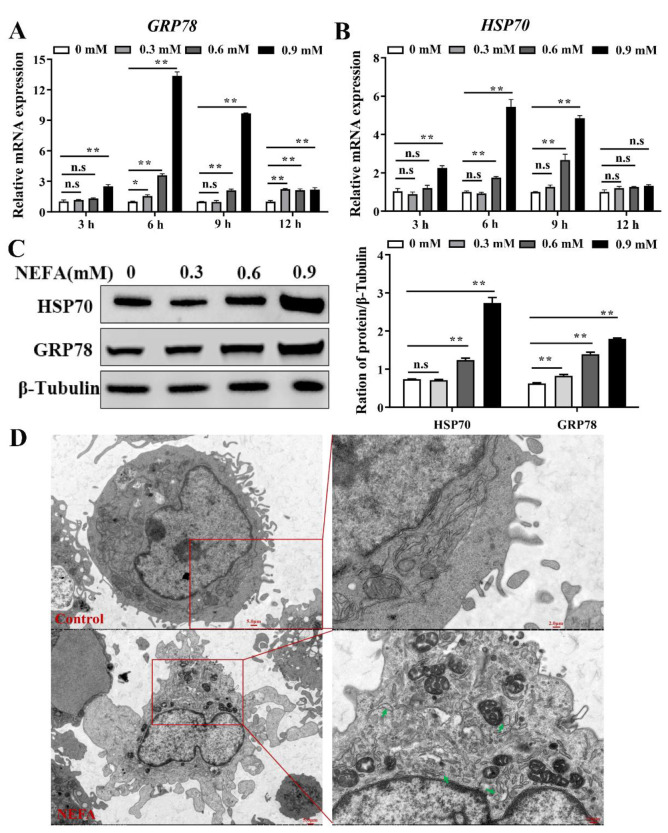
NEFA induced ER stress in bovine mammary epithelial cells. (**A**,**B**) Relative mRNA expression of GRP78 and HSP70 in BMECs after treatment with various concentrations (0.3, 0.6, and 0.9 mM) of NEFA for 3, 6, 9, and 12 h. (**C**) Protein expression of GRP78 and HSP70 in BMECs stimulated with the different concentrations of NEFA for 6 h. (**D**) TEM images of BMECs in the absence and presence of 0.9 mM NEFA for 6 h; green arrowheads represent ER expansion. Data are presented as the means ± the standard errors of the mean (SEM) of three independent experiments. n.s. means not significant, * *p* < 0.05; ** *p* < 0.01.

**Figure 2 metabolites-12-00803-f002:**
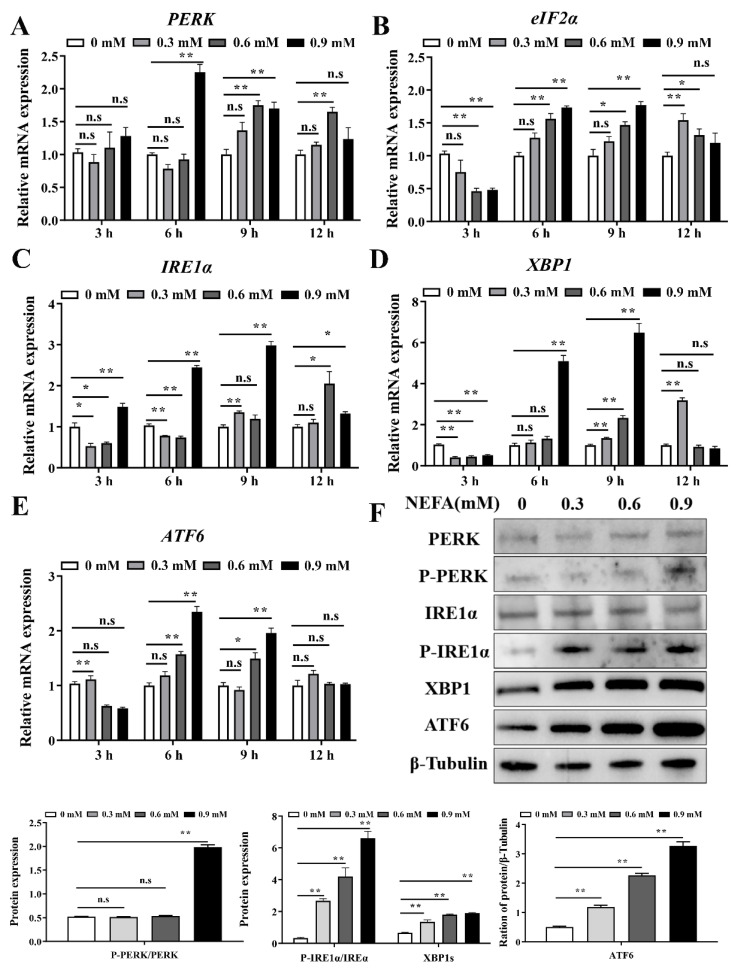
NEFA activated the three UPR signaling pathways in bovine mammary epithelial cells. (**A**–**E**) Relative mRNA expression of PERK, eIF2α, IRE1α, XBP1, and ATF6 in BMECs after treatment with various concentrations (0.3, 0.6, and 0.9 mM) of NEFA for 3, 6, 9, and 12 h. (**F**) Protein expression of PERK, P-PERK, IRE1α, P-IRE1α, XBP1, and ATF6 in BMECs stimulated with the different concentrations of NEFA for 6 h. Data are presented as the means ± the standard errors of the mean (SEM) of three independent experiments. n.s. means not significant, * *p* < 0.05; ** *p* < 0.01.

**Figure 3 metabolites-12-00803-f003:**
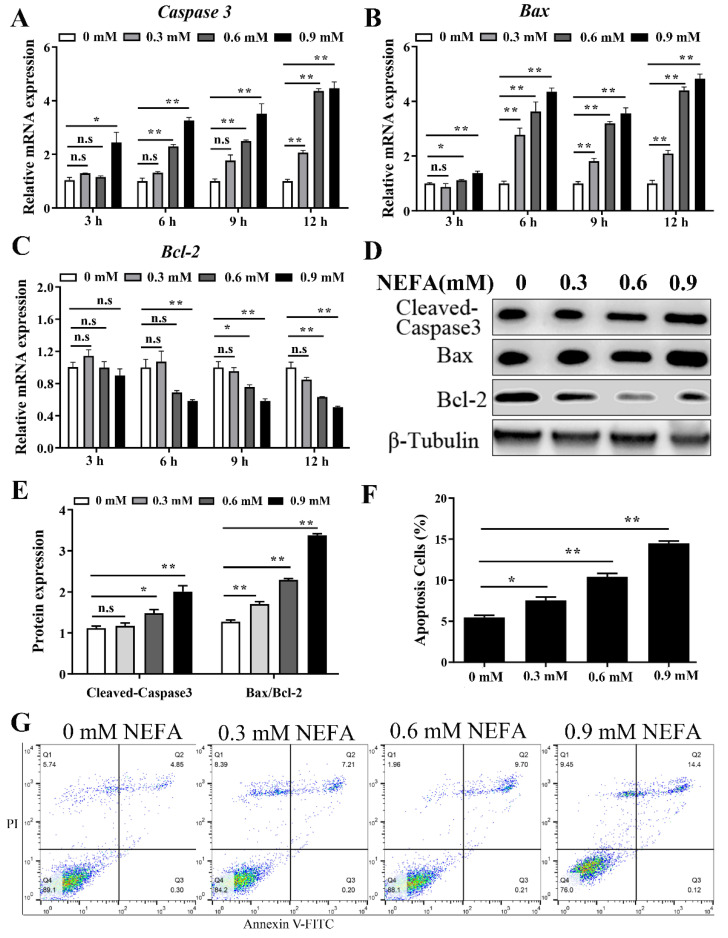
NEFA induced apoptosis in BMECs. (**A**–**C**) Relative mRNA expression of Caspase3, Bax, and Bcl2 in BMECs after treatment with various concentrations (0.3, 0.6, and 0.9 mM) of NEFA for 3, 6, 9, and 12 h. (**D**,**E**) Protein expression of Caspase3, Bax, and Bcl2 in BMECs stimulated with the different concentrate of NEFA for 6 h. (**F**,**G**) Cell apoptosis detected by flow cytometry using annexin V FITC/PI double staining. Data are presented as the means ± the standard errors of the mean (SEM) of three independent experiments. * *p* < 0.05; ** *p* < 0.01.

**Figure 4 metabolites-12-00803-f004:**
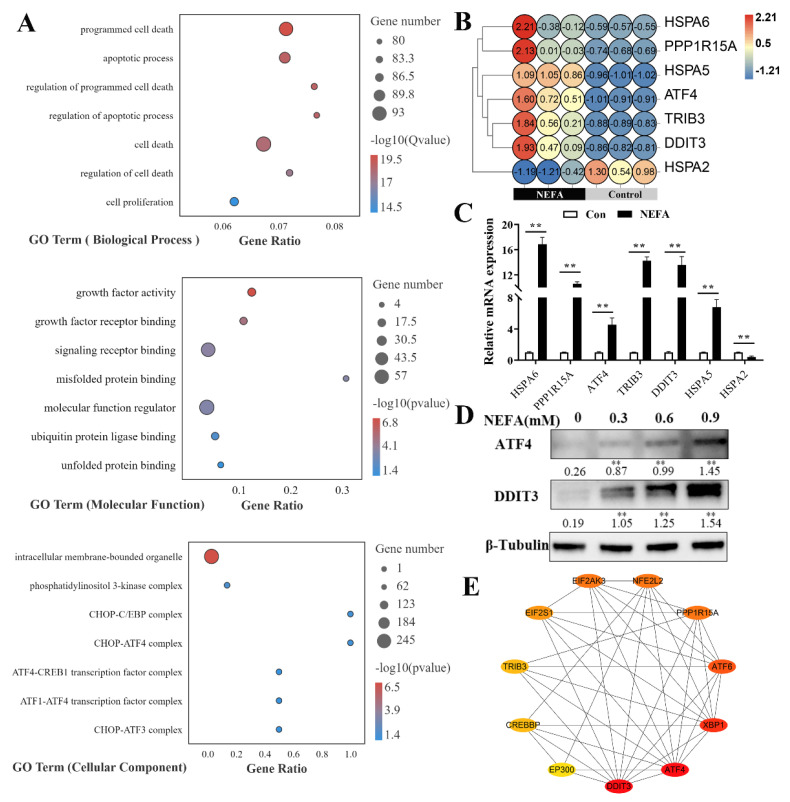
GO enrichment analysis and verification of the different NEFA treatment−activated genes. (**A**) Bubble plot of GO enrichment analysis. (**B**) Thermogram analysis of expressions (log10(FPKM+1)) of 7 target genes. (**C**) qPCR verification of expression modes of 7 target genes. (**D**) Protein expression of ATF4 and CHOP(DDIT3) in BMECs stimulated with NEFA for 6 h. (**E**) Protein and protein interaction network of CHOP(DDIT3) and ATF4. Data are presented as the means ± the standard errors of the mean (SEM) of three independent experiments. ** *p* < 0.01.

**Figure 5 metabolites-12-00803-f005:**
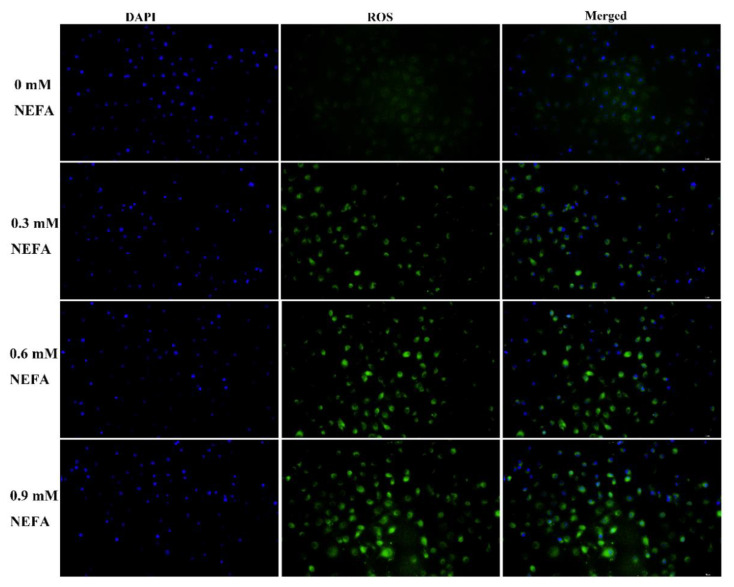
Accumulation of ROS in BMECs after NEFA treatment. Intracellular ROS levels in BMECs treated with different concentrations of NEFA (0, 0.3, 0.6, 0.9 mM) were observed by immunofluorescence (green); DAPI was used for nuclear staining (blue).

**Figure 6 metabolites-12-00803-f006:**
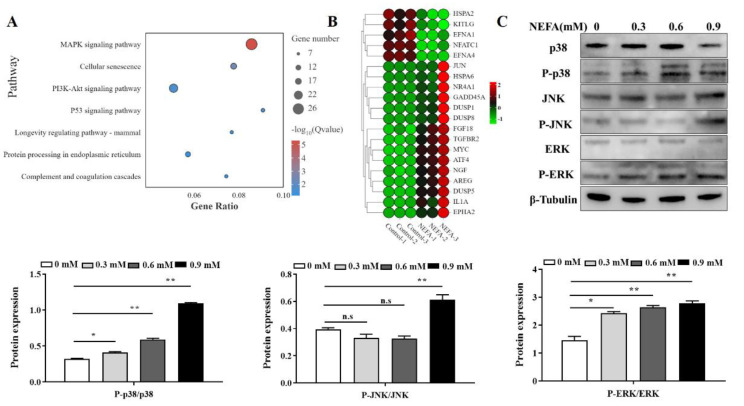
The MAPK signaling pathway was involved in NEFA−induced apoptosis. (**A**) The KEGG pathway enrichment analysis of the different NEFA treatment−activated genes. (**B**) Thermogram analysis of expressions of MAPK pathway−related genes. (**C**) The protein expression of MAPK pathway−related genes in BMECs stimulated with NEFA for 6 h. Data are presented as the means ± the standard errors of the mean (SEM) of three independent experiments. * *p* < 0.05; ** *p* < 0.01.

**Figure 7 metabolites-12-00803-f007:**
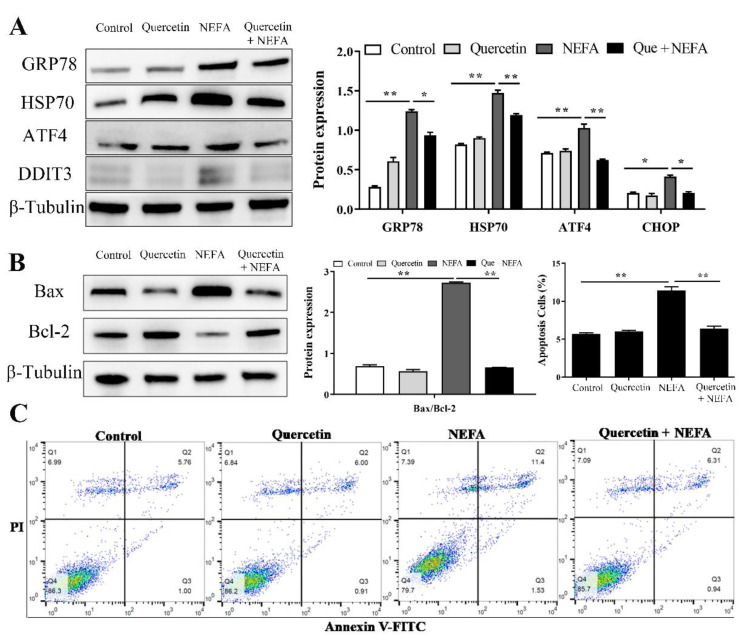
Quercetin alleviated ER stress-mediated apoptosis in NEFA-treated BMECs. Cells were pre-treated with quercetin (10 μM) for 24 h, then treated with 0.9 mM NEFA for 6 h. (**A**) Protein expression of ER stress marker proteins GRP78, HSP70, ATF4, and CHOP. (**B**) Protein expression of apoptosis marker proteins Bax and Bcl2. (**C**) Cell apoptosis detected by flow cytometry using annexin V FITC/PI double staining. Data are presented as the means ± the standard errors of the mean (SEM) of three independent experiments. * *p* < 0.05; ** *p* < 0.01.

**Figure 8 metabolites-12-00803-f008:**
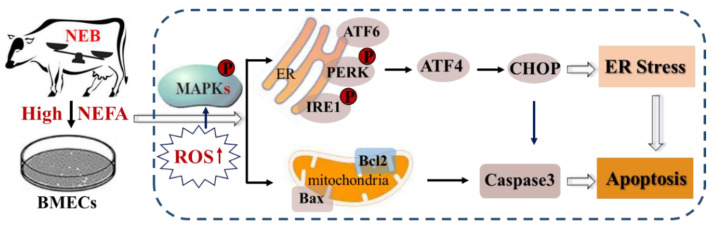
Signal activation pathways in BMECs treated with NEFAs. NEFA induces ROS accumulation, MAPKs activation, ER stress, and apoptosis in BMECs.

**Table 1 metabolites-12-00803-t001:** Primers sequences for real-time quantitative PCR analysis.

^a^ Gene	Accession Number	Primer Sequences (5′-3′)	(%)PCR Efficiency	*R* ^2^	Fragment Size (bp)
GRP78	NM_001075148.1	F: TCCAGGCTGGTGTGCTCTCT R: TCGTCAGGGGTCGTTCACCT	94	0.9977	222
DDIT3	NM_001078163.1	F: TCGGGGCACCTGTGTTTCAC R: CTCTGGTGGTCCTGGCTCCT	96	0.9981	112
PPP1R15A	NM_001046178.2	F: CTGATGGGCCTTCTCAGCCG R: TGCTTGGCTTCCAGGTTGGG	95	0.9958	248
HSPA2	NM_174344.1	F: GGCCATGAACCCCACCAACA R: ACTCCACCTGCACTTTGGGC	93	0.9966	143
TRIB3	NM_001076103.1	F: CCCCTGGCTGTTCCAGCAAA R: GGGCCCACTTCGAGCTTGTT	102	0.9992	105
ATF4	NM_001034342.2	F: TTGGGGGCTGAAGAGAGCCT R: CAGCCATTCGGAGGAGCCTG	95	0.9975	114
HSPA6	XM_002685850.5	F: AGGCCCAGAGAGACAGGGTG R: CCACAACTGCTGCCCACAGA	94	0.9968	280
PERK	NM_001098086.1	F: ACATGCTGTCCCCATCCCCT R: ATTGCTGGGCAAAGGGCTGT	101	0.9956	182
EIF2α	NM_175813.2	F: CATGCGCAGTGTGGTCAAGC R: CACAACAAGGTCCCACGCCA	94	0.9962	110
IRE-1α	XM_024980954.1	F: ACTGAGAGGGACCGGCAGTT R: TGGTCTGTTGCAGCAGGGTG	95	0.9979	127
XBP1	NM_001034727.3	F: GACCAAGGGGAATGGAGCGG R: GAAGGGGAGGCCGGTAAGGA	95	0.9984	290
ATF6α	XM_024989877.1	F: TGTGAGGGGAAGTAGGGGGC R: GGAGTGGTCCCCTGGGAAGT	93	0.9963	175
GAPDH	NM_001034034.2	F: CATGACCACTTTGGCATCGT R: CCATCCACAGTCTTCTGGGT	96	0.9985	133

^a^ GRP78, glucose-regulated protein 78; DDIT3, DNA damage inducible transcript 3; PPP1R15A, protein phosphatase 1 regulatory subunit 15A; HSPA2, heat shock protein family A (Hsp70) member 2; TRIB3, tribbles pseudokinase 3; ATF4, activating transcription factor 4; HSPA6, heat shock protein family A (Hsp70) member 6; PERK, eukaryotic translation initiation factor 2 alpha kinase 3; EIF2α, eukaryotic translation initiation factor 2 subunit alpha; IRE-1α, endoplasmic reticulum to nucleus signaling 1; XBP1, X-box binding protein 1; ATF6α, activating transcription factor 6; GAPDH, gallus glyceraldehyde-3-phosphate dehydrogenase.

## Data Availability

The authors confirm that all data underlying the findings are fully available, the relevant data are within the paper.

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
