# Peer review of "Non-Esterified Fatty Acid Induces ER Stress-Mediated Apoptosis via ROS/MAPK Signaling Pathway in Bovine Mammary Epithelial Cells"

_metabolites, 2022, doi:10.3390/metabo12090803_

Round 1
Reviewer 1 Report
In this paper, bovine mammary epithelial cells (BMECs) treated with different concentrations of NEFA were used to detect the expression of endoplasmic reticulum stress marker (UPR) and apoptosis signal related genes. The genes activated by NEFA treatment were analyzed by RNA-Seq. Those results showed that high levels of NEFA increased endoplasmic reticulum stress and promoted apoptosis of BMECs. In addition, it was found that MAPK signaling pathway was activated by the accumulation of reactive oxygen species. Quercetin can attenuate ER stress-mediated apoptosis in NEFA treated BMECs. The results of this study will promote the new exploration of the adaptive physiological mechanism of transitional cows to prevent or alleviate negative energy balance, and evaluate the potential application of quercetin as a new method to treat NEB in transitional cows. The following problems need to be solved before the article is published.
Major comment
1. In the author's experiment, there are many trends of correlation, but there is a lack of evidence of reverse genetics to prove the conclusion. It is suggested to supplement relevant experiments or modify the conclusions more appropriately.
2. In the process of cell culture, whether the production of intracellular NEFA in cells is considered, or whether different concentrations of NEFA will induce cells to produce different concentrations of intracellular NEFA?
3. How does transcriptome analysis screen for makp signals? The results are not shown. It is suggested to supplement the basis for screening MAPK.
Minor comment
1. Line91, ROS has many different components. What type of ROS does this kit detect?
2. Line148-150, after 6 and 9 hours of NEFA treatment with different concentrations (0.3, 0.6 and 0.9 mm) , there will be significant differences in the expression of stress markers. What is the reason for no significant difference after 12 hours?
3. Line182-183, lacks a description of the results of Fig (3F-3G).
4. Line216 should add references.
5. Line238-239, “treated with 0.9mm NEFA for 6 hours”, and Line72, in materials and methods the NEFA group is treated with 0.9mm for 4 hours?
6. In Figure 5, in DAPI staining and ROS fluorescence, there are many places with ROS fluorescence, but there is no DAPI signal. If this place is where the cells are located, it is suggested to supplement the cell photos under DIC.
7. In Figure 8, the names of ER and mitochondria should be marked.
Reviewer 2 Report
REVIEW
for the journal Metabolites (ISSN 2218-1989)
Article “Non-esterified Fatty Acid Induces ER Stress-Mediated Apoptosis via ROS/MAPK Signaling Pathway in Bovine Mammary Epithelial Cells”
Manuscript ID: metabolites-1838832
Authors: Yexiao Yan, Junpeng Huang, Changchao Huan, Lian Li, Chengmin Li
1. The effect of non-esterified fatty acid (NEFA) on bovine mammary epithelial cells (BMECs) has not been sufficiently studied to date. The study conducted by the authors is relevant both from a theoretical and practical point of view.
2. Line 32 “physiologi¬cal”. You need to delete the space.
3. Line 35. transi¬tion? Same fix.
4. Line 133: “All data are presented as mean ± SEM…”. It is necessary to explain to the reader the abbreviation SEM.
5. Line 132-1136: I did not find a more detailed description of statistical methods in terms of study design and sample sizes.
6. In the Materials and Methods section, I missed a detailed description of the study design and sampling.
7. Lines 139-140: “To determine the effect of NEFA on ER stress in bovine mammary epithelial cells, the expression of ER stress markers was tested after cells were treated with various concentrations (0, 0.3, 0.6, and 0.9 mM) of NEFA for 3, 6, 9, and 12 h by RT-qPCR”. In my opinion, the first sentence in the results section reflects the methodological points and should be in the methodology section. In addition, ow to understand the concentration of "0" for the reader?
8. Lines 217-218: “The correlation between accumulation of ROS and NEFA in BMECs was also studied.” The statistical analysis section does not state that you used correlation analysis and with what method?
9. Line 219. Figure 5 must be described.
10. Lines 221-223. Statements from the literature sources should be analyzed in the discussion section.
11. Lines 236-237. Same.
12. Line 254. Figure 8 must be described.
13. The list of references must be reviewed in detail and submitted according to the requirements. Pay attention to the spaces between the words.
14. The article is interesting, but the adjustments mentioned are recommended.
Sincerely, reviewer.
Reviewer 3 Report
The manuscript ID metabolites-1838832 entitled "Non-esterified Fatty Acid Induces ER Stress-Mediated Apoptosis via ROS/MAPK Signaling Pathway in Bovine Mammary Epithelial Cells" by Yexiao Yan et al. suggested that NEFA activate the ROS/MAPK signaling pathway to induce ER stress-mediated apoptosis, quercetin has been shown to alleviate ER stress-mediated apoptosis in NEFA treated BMECs. It was very interesting point of your observation and basic study, however, there are many points to be revised in this manuscript.

Round 2
Reviewer 1 Report
The authors have revised the manuscript according to the reviewer's comments.
Author Response
Thanks for your patience and the valuable comments.
Reviewer 2 Report
The authors took into account the recommendations presented. My suggestion: Accept in present form.
Author Response

(The authors gave the same response as above.)

Reviewer 3 Report
The manuscript ID metabolites-1838832 entitled "Non-esterified Fatty Acid Induces ER Stress-Mediated Apoptosis in Bovine Mammary Epithelial Cells Probably via ROS/MAPK Signaling Pathway" by Yexiao Yan et al. suggested that NEFA activate the ROS/MAPK signaling pathway to induce ER stress-mediated apoptosis, quercetin has been shown to alleviate ER stress-mediated apoptosis in NEFA treated BMECs. It was very interesting point of your observation and basic study. The authors revised well, however, there are some points to be revised in this manuscript. In addition, in this Journal “metabolites”, please check the necessary to deposit the sequence data from RNA-seq in DNA Data Bank before submitting the paper.

Author Response
Dear Reviewer:
Thanks for your patience and the valuable comments. We have studied your comments carefully and have revised our manuscript according to the comments. Our responses to the comments are as follows. The changes in the revised manuscript have been highlighted using yellow highlighting. Hopefully we have addressed all of your concerns.
Thank you once again!
Best regards,
Chengmin Li
School of Biotechnology, Jiangsu University of Science and Technology
E-mail address: chengminli@just.edu.cn
Comments to the Authors
The manuscript ID metabolites-1838832 entitled "Non-esterified Fatty Acid Induces ER Stress-Mediated Apoptosis in Bovine Mammary Epithelial Cells Probably via ROS/MAPK Signaling Pathway" by Yexiao Yan et al. suggested that NEFA activate the ROS/MAPK signaling pathway to induce ER stress-mediated apoptosis, quercetin has been shown to alleviate ER stress-mediated apoptosis in NEFA treated BMECs. It was very interesting point of your observation and basic study. The authors revised well, however, there are some points to be revised in this manuscript. In addition, in this Journal “metabolites”, please check the necessary to deposit the sequence data from RNA-seq in DNA Data Bank before submitting the paper
INTRODUCTION:
L61: bovine mammary epithelial cells (BMECs), because you already defined BMECs in L45.
Response: Many thanks for this comment. Change has been made in line 60 of revised manuscript.
MATERIALS AND METHODS:
L70-72: The media was composed DMEM and 10% FBS only? Was this true? No antibiotic and hormones?
Response: Thank you for reminding us of this and we are really sorry for the unclear description. Here we used the Dulbecco’s Modified Eagle’s Medium (DMEM)/F12 medium with 10% FBS for the cell culture, which can meet the needs of cell growth. The information on DMEM/F12 was supplemented in our revised manuscript (please see line 70-71). Since BMEC cell line (MAC-T) are less prone to contamination, no antibiotics were added.
L120-122 and Table 1: IMPORTANT! The reviewer cannot judge your ddCt method without PCR Eff%, R2 value of each primer.
Response: Thanks for your comments and patience. The PCR Eff% and R2 value of each primer were added in the revised Table1.
L132-151: You need the information about RNA-seq; What the bovine reference genome? Ensembl_release104? I am not sure you did not need to deposit the sequence data from RNA-seq in DNA Data Bank before submitting the paper in “metabolites”.
Response: Thanks for your comments and patience. We are awfully sorry for the inappropriate revision. The bovine reference genome is “Bos Taurus, assembly ARS-UCD1.2 (GCA_002263795.2),http://asia.ensembl.org/Bos_taurus/Info/Index”. And the raw data were submitted to the National Center for Biotechnology Information (NCBI) under BioProject accession number PRJNA869860. We have added the information in the revised manuscript (please see line 143-144, 153-154).
RESULTS
L176-178: The sentence was not adequate. The GRP78 expression level in NEFA concentrations 0.3 mM for 9h was not significantly different compared to the control.
Response: Thank you for pointing this out to us, and we are awfully sorry for the inadequate description. Changes have been made in line 180-182 of the revised manuscript.
L179-183: The sentence was not adequate. The HSP70 expression levels in NEFA concentrations 0.3 mM for 3, 6, and 9h were not significantly different compared to the control.
Response: Thanks for your comments and patience. Changes have been made in line 184 of the revised manuscript.
L190: NEFA for 3, 6, 9 and 12 h.
Response: Thank you for reminding us of this. Changes have been made in line 193 of the revised manuscript.
L208-210: The sentence was not adequate. The Caspase3 expression levels in NEFA concentrations 0.3 mM for 3, 6, and 9h were not significantly different compared to the control.
Response: We are sorry for the inadequate description. Changes have been made in line 211-213 of the revised manuscript.
Fig. 8: The position of effect of Quercetin is right? Quercetin→ROS→MAPKs??? This study cannot demonstrate clearly the point of effect of Quercetin.
Response: Thanks for your comments and patience. we agree with the reviewer’s comment. Further study will need to be done to ascertain the mechanisms of quercetin on ER stress-mediated apoptosis in NEFA treated BMECs. Changes have been made in the revised Figure 8.
DISCUSSION
L295, 332-333: BMECs
Response: Many thanks for this comment. Change has been made in line 297 of revised manuscript.
L352: as evidenced by the upregulated expression of ER stress and apoptosis markers (Figure 7)? Is this true? I think that evidenced by the downregulated expression of ER stress and apoptosis markers (Figure 7)
Response: Thank you very much for your constructive comments and sorry for the confusion caused by my inadequate expression. We agree with the reviewer’s comment and we have changed the expression in line 354 of the revised manuscript.

Round 3
Reviewer 3 Report
I have no more comments. Thank you.